# Minimal Residual Disease Significance in Multiple Myeloma Patients Treated with Anti-CD38 Monoclonal Antibodies

**DOI:** 10.3390/ph18020159

**Published:** 2025-01-25

**Authors:** Federico Caroni, Vincenzo Sammartano, Paola Pacelli, Anna Sicuranza, Margherita Malchiodi, Andreea Dragomir, Sara Ciofini, Donatella Raspadori, Monica Bocchia, Alessandro Gozzetti

**Affiliations:** AOUS Policlinico Le Scotte, University of Siena, 53100 Siena, Italy; federicocaroni1@gmail.com (F.C.); sammartano2@student.unisi.it (V.S.); paola.pacelli@unisi.it (P.P.); sicuranza4@unisi.it (A.S.); m.malchiodi@student.unisi.it (M.M.); a.dragomir@student.unisi.it (A.D.); sara.ciofini@ao-siena.toscana.it (S.C.); raspadori@unisi.it (D.R.); bocchia@unisi.it (M.B.)

**Keywords:** multiple myeloma, minimal residual disease, monoclonal antibodies

## Abstract

Minimal residual disease (MRD) evaluation is a recognized endpoint in clinical trials. Both next-generation flow and sequencing could be used as complementary techniques to detect myeloma cells after therapy to measure the depth of response and novel drug efficacy. Anti-CD38 monoclonal antibodies combined with proteasome inhibitors and immunomodulatory drugs have increased the quality of response in myeloma patients, and MRD evaluation is also entering routine clinical practice in many hematological centers. This review analyzes updated results from recent clinical trials utilizing anti-CD38 monoclonal antibodies such as isatuximab and daratumumab in terms of their responses and MRD data. MRD-driven therapy appears promising for the future of MM patients, and emerging minimally invasive techniques to assess MRD are under investigation as novel potential methods to replace or integrate traditional MRD evaluation.

## 1. Introduction

Multiple myeloma (MM) is the most common hematological disease after NHL [1]. MM is characterized by monoclonal plasma cells in the bone marrow derived from a precursor cell that accumulated genomic aberrations and that produces a monoclonal component in the serum or urine that can lead to hypercalcemia, renal failure, lytic bone lesions, and anemia [2,3,4]. The microenvironment has been shown to be important, and even a small clone could be sufficient to produce damage [5,6,7]. The disease has often been defined as incurable, but in the last two decades, many advances have been made in progression-free survival (PFS) and overall survival (OS) due to novel drugs and novel strategies for cures. [8,9,10,11]. OS has increased from a median of 3–4 years to a median of 8–9 years [12,13]. Before the early 2000s, the only drugs used were chemotherapeutic agents [14,15,16,17,18]. The advent of thalidomide as the first immunomodulatory drug (IMID) [19,20] and then the first proteasome inhibitor (PI) bortezomib [21,22,23] changed the landscape of the disease and led to a rapid escalation of progress with lenalidomide and pomalidomide; the proteasome inhibitors carfilzomib and ixazomib; and the monoclonal antibodies (MoA) daratumumab, elotuzumab, isatuximab, and belantamab [24,25,26,27,28,29,30,31,32,33,34,35]. These drugs have been used alone or in combination with others in triplets and quadruplets, with high and intense responses, and also in aggressive high-risk conditions, such as extramedullary diseases and those with high-risk genetics [36,37,38,39,40,41,42,43,44,45,46]. Moreover, MoA directed towards CD38 led to high rates of complete remission and an increasing depth of response [39,47,48,49,50,51,52,53,54,55].

Since response rates have increased, MRD was introduced by the International Myeloma Working Group (IMWG) for response evaluation. MRD can be defined as the persistence of myeloma cells below the threshold of detection when using conventional morphologic methods. Multi-flow cytometry, i.e., next-generation flow (NGF) and next-generation sequencing (NGS), can be utilized to detect MRD with a sensitivity of 10^−6^ in the bone marrow aspirate (Figure 1) [4]. MRD negativity is related to a better survival, and it is mandatory in clinical trials while it is still not widely used in clinical practice [56,57]. The rate of MRD negativity reached by combination therapies including anti-CD38 MoA daratumumab and isatuximab has been unprecedented in multiple myeloma. This review will focus on the latest results of clinical trials of these two drugs.

CD38 seems to be a good target in MM since transformed plasma cells express it at higher levels compared with normal cells [24,25,26,27,28,29,30,31,32,33,34,35]. Daratumumab is the first anti-CD38 MoA and is currently used in several combination therapies in both newly diagnosed MM (NDMM) and relapsed–refractory MM (RRMM) settings. The second-generation anti-CD38 MoA isatuximab is approved by EMA only for lenalidomide-refractory patients. Currently, daratumumab can be administered by intravenous infusion or subcutaneously; conversely, isatuximab can be given only by intravenous infusion. The combination of daratumumab, bortezomib, thalidomide, and dexamethasone is approved for NDMM patients who are eligible for autologous stem cell transplantation (ASCT). For transplant-ineligible (TIE) MM patients, the triplet daratumumab, lenalidomide, and dexamethasone and the quadruplet daratumumab, bortezomib, melphalan, and dexamethasone can be used as first-line therapies. In RRMM, both daratumumab, bortezomib, and dexamethasone and daratumumab, lenalidomide, and dexamethasone can be used as the second line of treatment. Also, daratumumab, pomalidomide, and dexamethasone are approved for patients previously treated with lenalidomide and a PI. Isatuximab is approved for the second line in combination with carfilzomib and dexamethasone or for the third line with pomalidomide and dexamethasone.

## 2. MRD Techniques and Applications

### 2.1. The Role of NGS and NGF in MM MRD Evaluation

Nowadays, the gold standards for MRD in MM are represented by the two main high-throughput techniques, multiparametric flow cytometry (MFC) and NGS [59]. MFC can efficiently detect MM plasma cells (PCs) via the expression of aberrant markers such as CD56, CD28, and CD117 and the lack of CD45, CD19, CD27, and CD81. These markers, with the clonal restriction of MM PCs to just one of two immunoglobulin light chains, κ or ʎ, contribute to distinguishing normal from clonal MM PCs. The advances achieved by MFC thanks to the work of the EuroFlow Consortium and the development of NGF, a highly standardized approach based on the use of certified antibody panels, associated with the use of Standard Operational Procedures [60,61], lead to a high sensitivity of 10^−5^ or 10^−6^ that is comparable and superimposable to NGS assays (Figure 1) [58]. NGF advantages reside in the high applicability (almost 100% of MM cases) and low time consumption (3–4 h of processing) of this technique; furthermore, it does not require a diagnostic sample to be applied [62]. Some minor biases are the need to work on fresh samples and the necessity to acquire at least 10^7^ cells per sample to be able to reach the desired and relevant sensitivity levels, expressed as the limit of quantification and the limit of detection [59]. On the other hand, NGS technology, through the parallel sequencing of millions of reads, is considered a valid molecular method allowing for MRD measurement with 10^−6^ sensitivity [63]. The advantage of this approach resides in the ability to identify clonality in MM patients with a low tumor burden, without designing patient-specific primers or a standard curve for “MRD quantification”. Two identical sequencing reads were defined as clonotypes, and the >5% frequency of a clonotype, established as the cut-off value for MRD evaluation in follow-up samples, is referred to as clonality. However, the great amount of data produced needs to be elaborated through specific bioinformatic tools designed to analyze millions of reads. Several NGS platforms for MRD detection in MM have been tested; the ClonoSEQ^®^ (Adaptive Biotechnologies, Seattle, WA, USA) was the first to be licensed by the FDA in 2019, and it is currently the most frequently adopted. Despite the high sensitivity of NGS, the feasibility of this approach is limited by high costs, long turnaround times, and the expertise required for data analysis. More affordable techniques, such as digital PCR, were proposed to be used in daily MM patient management as it does not require a standard curve and is less laborious than NGS regarding data interpretation. On the other hand, its applicability is not yet standardized [64,65]. The IMWG recommendation is to choose NGF or NGS based on local availability. The combination of the two techniques may offer a chance for a better follow-up of MM patients and cover all the possible pitfalls associated with monitoring MM MRD [4].

### 2.2. MRD Initial Experience in Clinical Trials

The first piece of evidence on the prognostic value of MRD negativity came from a pooled analysis of three PETHEMA/GEM clinical trials of both transplant-eligible (TE) and TIE MM patients who had MRD assessments 9 months after study enrollment [66]. Generally, patients achieving a complete response (CR) had better survival outcomes than patients with near-CRs (i.e., negative serum protein electrophoresis and positive immunofixation) and partial responses (PRs) or less; however, the real benefit in terms of PFS and OS was observed in MRD-negative patients. Indeed, patients achieving CRs but positive MRD had PFS and OS like the ones who achieved near-CRs or PRs. In 2020, a large meta-analysis of several randomized and observational clinical trials included more than 8000 MM patients, and prolonged PFS and OS were demonstrated for patients achieving MRD negativity in comparison with MRD-positive patients [67]. Furthermore, MRD negativity was an independent positive prognostic factor, since survival outcomes were better in MRD-negative patients in several subgroups, including TE or TIE NDMM RRMM, having a cytogenetic standard risk or high-risk disease (the latter defined as the presence of t(4;14), t(14;16), or del(17p)) ≥ CR or ≤ very good partial response (VGPR).

In the NDMM setting, the IFM 2009 trial compared bortezomib, lenalidomide, and dexamethasone (VRd) alone for eight cycles to VRd induction for three cycles followed by ASCT, two consolidation VRd courses, and lenalidomide maintenance for 12 months [68]. With a median follow-up of 93 months, the early transplant group showed a significant prolonged PFS compared to patients treated with VRd alone (47 vs. 35 months). Interestingly, patients achieving MRD negativity had longer PFS and OS compared to the MRD-positive ones, independently from the treatment arm.

### 2.3. MRD and Trials with Anti-CD38 Monoclonal Antibodies

Anti-CD38 MM trials are summarized in Table 1, which specifies the population settings and the treatment regimens and highlights the MRD data of each trial.

#### 2.3.1. Transplant-Eligible (TE) MM Patients

In the phase 3 CASSIOPEIA trial, TE NDMM patients were randomized to bortezomib, thalidomide, and dexamethasone (VTd) alone or in combination with daratumumab (Dara-VTd) [69]. The rate of stringent complete response (sCR) 100 days after ASCT was 29% for patients treated with daratumumab vs. 20% in the VTd group and the rate of MRD negativity with Dara-VTd, evaluated using NGF cytometry, was 64% and 57% using NGS, as compared to 44% and 37%, respectively, in the VTd arm.

Daratumumab was also combined with VRd in the phase 3 PERSEUS trial, in which TE patients with NDMM received either daratumumab combined with VRd induction and consolidation therapy and with lenalidomide maintenance therapy (Dara-VRd group) or VRd induction and consolidation followed by lenalidomide maintenance alone (VRd group) [70]. The estimated percentage of patients with PFS at 48 months was 84.3% in the Dara-VRd group and 67.7% in the VRd group; MRD negativity was observed in 75.2% of the Dara-VRd group vs. 47.5% of the control group, respectively.

TE NDMM patients enrolled in the phase 2 GRIFFIN trial were randomly assigned to the Dara-VRd or VRd induction treatment, followed by ASCT and subsequent Dara-VRd or VRd consolidation and lenalidomide with or without daratumumab maintenance therapy for 2 years [51]. The addition of daratumumab to the therapeutic scheme improved the rates of sCR at the end of consolidation and response rates intensified over time (with a median follow-up of 49 months, sCR rates were 67% in Dara-VRd vs. 48% in VRd). The rate of MRD negativity was also higher in the daratumumab-containing arm, increasing over time from post-induction (21% vs. 6%) to the end of consolidation (47% vs. 16%) and until the end of 24 months of maintenance (64% vs. 30%). These findings were associated with a prolonged PFS in patients treated with Dara-VRd, with a 4-year PFS of 87% compared to 70% in the VRd arm. MRD-negativity rates were higher in the Dara-VRd arm in all the subgroups analyzed. Furthermore, the rate of sustained MRD negativity lasting ≥12 months was higher in the Dara-VRd group (44% vs. 12%).

The use of daratumumab in combination with carfilzomib, lenalidomide, and dexamethasone (Dara-KRd) has been explored in the phase 2 MASTER trial for TE NDMM patients; of note, in this trial, 57% of patients had a high-risk cytogenetic disease [71]. The trial plan established that patients who achieved two consecutive assessments of MRD negativity interrupted the treatment, whereas those who did not were assigned to lenalidomide as maintenance therapy. With a median follow-up of 42 months, the 3-year PFS was 77% with a relevant inferior outcome for patients with ≥2 high-risk cytogenetic abnormalities (HRCAs) in comparison to standard risk patients or those with only 1 HRCA (50% vs. 88% vs. 79%). The rate of MRD negativity, evaluated through NGS with a sensitivity of 10^−5^, was 42% following induction therapy and increased up to 81% at the end of consolidation. Moreover, maintenance was interrupted in 71% of patients and, among these patients, 73% were still treatment-free and MRD-negative at the last follow-up. No significant differences in the rate of MRD negativity have been found among patients with 0, 1, or ≥2 HRCAs; however, patients with ≥2 HRCAs had higher cumulative rates of progression two years after the discontinuation of the treatment in comparison to standard risk patients or those with only 1 HRCA (47% vs. 9% vs. 9%). All these data suggest that there is still an unmet need for ultra-high-risk MM, despite the achievement of intense responses, including undetectable MRD, with quadruplet treatment regimens [37]. Similarly, the phase 2 LCI-HEM-MYE-KRdD-001 trial for NDMM patients evaluated the efficacy of Dara-KRd induction followed by an MRD-adapted strategy [72]. MRD-negative patients could receive lenalidomide maintenance or no further treatments, whereas MRD-positive patients underwent ASCT if they were TE or KRd consolidation if they were TIE. A total of 77.8% of patients receiving lenalidomide demonstrated sustained MRD negativity for 12 months, while 62.5% of TE MRD-positive patients converted to an MRD-negative status; conversely, none of the TIE MRD-positive patients achieved MRD negativity.

Induction therapy with isatuximab, bortezomib, lenalidomide, and dexamethasone (Isa-VRd) was investigated in the setting of TE NDMM patients in the phase 3 GMMG-HD7 trial, in which single or tandem ASCT after induction and maintenance with lenalidomide alone or isatuximab and lenalidomide for 3 years were given [73]. Higher MRD-negativity rates were reported with Isa-VRd compared to the control induction therapy with VRd, and MRD-negativity rates continued to increase after the transplant (66.2% vs. 47.7%). Patients treated with Isa-VRd had prolonged PFS, regardless of maintenance therapy.

#### 2.3.2. Transplant-Ineligible (TIE) MM Patients

The prognostic role of MRD has also been established in the setting of TIE NDMM patients in the two randomized prospective phase 3 clinical trials MAIA and ALCYONE. In the MAIA trial, patients aged more than 65 years were randomized to first-line therapy with lenalidomide and dexamethasone (Rd) with or without daratumumab until progressive disease (PD) [74]. With a median follow-up of 56 months, PFS was longer in the daratumumab group; indeed, median PFS was not reached in the daratumumab group vs. 34 months in the Rd group. MRD was evaluated in NGS, and higher rates of MRD negativity were observed for daratumumab-treated patients (31% vs. 10%); moreover, MRD-negativity rates increased over time for both arms, but in a more pronounced manner in the Dara-Rd group, and patients treated with Dara-Rd had a longer duration of MRD negativity (sustained MRD negativity lasting ≥6 months or ≥12 months was 15% and 11%, respectively; on the other hand, for the Rd group, MRD negativity lasting ≥6 months or ≥12 months was 4% and 2%) [88]. The ALCYONE trial compared the fixed-time triplet bortezomib, melphalan, and prednisone (VMP) alone or in combination with daratumumab followed by a monthly daratumumab maintenance until progression [75,89]. With a median follow-up of 40 months, the addition of daratumumab resulted in better survival outcomes, with a median PFS of 36 months and a 3-year OS of 78% (vs. 19 months and 68% for patients treated with VMP). Again, the MRD-negativity rate was higher in daratumumab-treated patients (22% vs. 6%), who also had a more durable MRD negativity compared to the control group (the rate of sustained MRD negativity lasting ≥6 months or ≥12 months in the Dara-VMP arm was 16% and 14%, respectively, in comparison with rates of 5% and 3% in the VMP arm). In both MAIA and ALCYONE trials, patients who achieved MRD negativity had a longer PFS as compared to the MRD-positive ones, and this advantage was independent of the treatment arm; in addition, patients achieving a sustained MRD negativity (lasting ≥6 months or ≥12 months) had longer PFS than patients with persistence of MRD positivity or whose MRD negativity lasted less than 6 months [90].

The phase 2 GMMG-CONCEPT trial investigated MRD in a specific cohort of high-risk NDMM patients, defined as ISS II or III and del(17p), t(4;14), or t(14;16) or gain(1q21) with ≥ 3 copies [76]. Both TE and TIE patients received isatuximab, carfilzomib, lenalidomide, and dexamethasone (Isa-KRd) induction and consolidation, followed by isatuximab, carfilzomib, and lenalidomide (Isa-KR) maintenance, achieving high rates of MRD negativity which translated into a median PFS not being reached. Overall, 81.8% of TE and 69.2% of TIE patients reached MRD negativity; 62.6% of TE and 46.2% of TIE patients had sustained MRD negativity for at least 12 months.

The efficacy of Isa-VRd in TIE NDMM patients was demonstrated in the phase 3 BENEFIT and IMROZ trials [77,78]. Patients enrolled in the BENEFIT trial were randomized to Isa-VRd or isatuximab, lenalidomide, and dexamethasone (Isa-Rd), and higher 18-month MRD-negativity rates were reported in the quadruplet therapy group (53% vs. 26%).

In the IMROZ trial, patients were randomized to the Isa-VRd or VRd control arm, and patients in the experimental treatment arm had significantly prolonged PFS (with a median follow-up of 59.7 months, median PFS was not reached in the isatuximab group vs. 54.3 months with the VRd regimen alone) without new concerning safety issues, leading to the recent FDA approval. Furthermore, more intense responses were observed in the Isa-VRd group with higher ≥ CR rates (74.7% vs. 64.1%), MRD-negativity rates ≥ CRs (55.5% vs. 40.9%), and ≥12 months of sustained MRD-negativity rates (46.8% vs. 24.3%).

#### 2.3.3. Consolidation/Maintenance After ASCT

The phase 3 AURIGA study compared daratumumab and lenalidomide maintenance therapy versus standard-of-care lenalidomide alone after ASCT [79]. The MRD negativity conversion rate at 12 months was 50% vs. 18% and PFS at 32 months was 82% vs. 66% in favor of the daratumumab group. In the phase 2 DART4MM trial, daratumumab was given as consolidation therapy in an MM population of patients already in >VGPR but MRD-positive at trial screening [80]. Fifty patients received daratumumab at the primary endpoint of 6 months and 30% converted to MRD negativity.

#### 2.3.4. Relapsed MM Patients

Outcome advantages in patients achieving MRD negativity were also observed in the setting of RRMM, as demonstrated by the phase 3 clinical trials POLLUX and CASTOR [81,82]. In these trials, patients treated with at least one prior line of therapy received Rd with or without daratumumab in the POLLUX trial or bortezomib and dexamethasone (Vd) alone or in combination with daratumumab (Dara-Vd) in the CASTOR trial. Both studies showed that the addition of daratumumab significantly reduced the risk of progression or death (>50%); moreover, the responses intensified over time, with MRD-negative rates becoming increasingly higher in the treatment arms including daratumumab. Furthermore, the rate of sustained MRD negativity was higher in the Dara-Rd group (MRD negativity lasting ≥6 months was 20% vs. 2% and MRD negativity lasting ≥12 months was 16% vs. 1%) and in the Dara-Vd group (MRD negativity lasting ≥6 months was 10% vs. 1% and MRD negativity lasting ≥12 months was 7% vs. 0%). Finally, the persistence of an MRD-negative status was associated with a prolonged PFS. In the phase 3 CANDOR trial, RRMM patients treated with one to three prior lines of therapy were randomized to carfilzomib and dexamethasone (Kd) or daratumumab, carfilzomib, and dexamethasone (Dara-Kd) [83]. Patients treated with Dara-Kd had higher PFS rates (28.4 vs. 15.2 months) and MRD–negativity rates (28% vs. 9%).

A combined analysis of all patients enrolled in POLLUX, CASTOR, MAIA, and ALCYONE trials was recently conducted to evaluate the rate of MRD negativity in a large cohort of patients enrolled in pivotal studies [88]. Overall, 17% of all patients achieved an MRD-negative status assessed by NGS with a threshold of 10^−5^ and the rate of MRD negativity was higher in the cohort of patients treated with daratumumab-containing regimens (27% vs. 6% in the control arms). Patients achieving CR and MRD negativity had a longer PFS as compared to the MRD-positive patients or those achieving no more than a VGPR, with an estimated 48-month PFS of 70% vs. 24%. More intense responses for patients treated with daratumumab were also observed in the cytogenetic high-risk subgroup, where 23% of patients achieved a CR and MRD-negative status (vs. 4% in the control groups).

The phase 3 APOLLO study compared daratumumab plus pomalidomide and dexamethasone (Dara-Pd) versus pomalidomide and dexamethasone (Pd) alone in previously treated MM [84]. MRD was assessed by NGS (at a threshold of one tumor cell per 10^5^ white cells) at the time of suspected CR or sCR; at 6, 12, 18, and 24 months; and every 12 months after achieving CR or sCR, until disease progression. An improved PFS was observed in the Dara-Pd group (12.4 months vs. 6.9 months) and the rate of ≥CR and negativity was higher in the Dara-Pd group than with the Pd regimen alone (CR or better: 25% vs. 4%; MRD negativity: 9% vs. 2%). Similarly, the phase 3 ICARIA-MM study evaluated the addition of isatuximab with pomalidomide and dexamethasone (Isa-Pd) versus Pd alone in patients with RRMM [85,86]. MRD was assessed by NGS (at the sensitivity level of 10^−5^) once patients achieved CR or sCR and 3 months later in case of positive MRD. The Isa-Pd group showed better OS and PFS compared to the control group and MRD negativity was observed in 5% of patients in the Isa-Pd group and in none of the patients in the Pd-alone group.

In the phase 3 IKEMA study, the efficacy of isatuximab plus carfilzomib and dexamethasone (Isa-Kd) was compared to Kd alone in RRMM [87]. Median PFS was not reached in the isatuximab group compared with 19.15 months in the control group. MRD was assessed by NGS with a minimum sensitivity of 10^−5^ nucleated cells in patients reaching VGPR or better on day 1 of every cycle and when treatment stopped. The MRD negativity rate in the Isa-Kd group was 30% (vs. 13% in the control group), which is an impressive result considering that patients had received a median of two previous lines of therapy.

## 3. Discussion and Future Directions in MRD for MM

Monoclonal antibodies changed the MM treatment, increasing disease disappearance in the bone marrow and the long-term survival of MM patients. Bone disease should also be evaluated at diagnosis and for confirming responses using metabolic imaging techniques [91,92,93,94,95,96,97,98,99]. Minimally invasive alternatives for MRD assessment in MM are also under investigation. Promising results for evaluating MM in peripheral blood from a liquid biopsy derived from novel techniques such as detections of circulating tumor cells (CTCs), identification of cell-free DNA, and measurement of monoclonal protein concentration with mass spectrometry (MS) [100,101]. One study demonstrated that positive MRD in PB via the detection of CTCs could be a surrogate of MRD in BM and represent an independent prognostic factor; however, this technique displays high rates of false negatives. Hence, additional studies are required to improve the diagnostic sensitivity of the detection of CTCs and to better define their role in MRD assessment [102]. Similarly, cell-free DNA for MRD evaluation in MM is still considered less sensitive than bone marrow-based techniques; however, its use may provide a prognostic role. Indeed, a recent study by Mithraprabhu et al. demonstrated that peripheral blood reduction in cell-free DNA burden in combination with negative MRD via NGF on bone marrow is associated with superior outcome in MM patients [103]. Eventually, MS will detect low levels of M protein produced by residual MM cells, representing an attractive and accurate alternative to bone marrow MRD evaluation [104]. Mai et al. proved that MM patients with negative MRD in the bone marrow and MS had favorable PFS indicating that MS could be combined with NGS or NGF MRD to improve its prognostic value [105]. Furthermore, a recent study confirmed that MS represents a highly sensitive and minimally invasive method of MRD monitoring in MM [106]. Hence, further studies are necessary to favor the highly anticipated integration of these techniques in clinical practice. Recent trials are also evaluating whether MRD analysis could be a useful biomarker for discontinuation of treatments. In the GEM2014MAIN clinical trial, Rd was compared to the triplet ixazomib, lenalidomide, and dexamethasone (Ixa-Rd) as maintenance therapies following VRd induction, as well as ASCT and VRd consolidation [107]. With an MRD-oriented approach, patients discontinued maintenance 2 years after its initiation if they were found to be MRD-negative, as assessed by NGF cytometry on bone marrow aspirate with a sensitivity of 3 × 10^6^; conversely, MRD-positive patients continued maintenance therapy for 3 additional years with the Rd (and reduced steroid doses). The addition of ixazomib to Rd did not prolong the PFS (6-year PFS rate of 61% for Rd vs. 55% for Ixa-Rd); instead, patients with undetectable MRD had a significantly longer PFS compared to the MRD-positive ones (5-year PFS 89% vs. 57%). Furthermore, patients with persistent MRD positivity or with MRD resurgence after prior MRD negativity had a reduced PFS compared to those with sustained MRD negativity or those who achieved MRD negativity later (66-month PFS: 54% vs. 89%). Interestingly, at the same time as bone marrow analyses, MRD was also studied via quantitative immunoprecipitation MS on peripheral blood, obtaining superimposable results to those of NGF cytometry. Three prognostic factors were found to predict MRD resurgence or PD: ISS 3, more than 0.01% of CTCs at diagnosis, and failure to achieve MRD negativity after induction [108]. The 5-year rates of MRD resurgence and/or PD were 16%, 33%, and 57% in patients having 0, 1, or ≥2 risk factors, respectively. Other ongoing trials, such as MIDAS, MASTER-2, RADAR, and ADVANCE, are exploring intensified or de-escalated therapies according to MRD status; some trials will also provide insights into the cessation of maintenance or consolidation therapies for MRD-negative patients, for example, MRD2STOP and DRAMMATIC studies [109]. The future seems brighter for MM patients: effective therapies and MRD negativity are important tools toward a cure.

## Figures and Tables

**Figure 1 pharmaceuticals-18-00159-f001:**
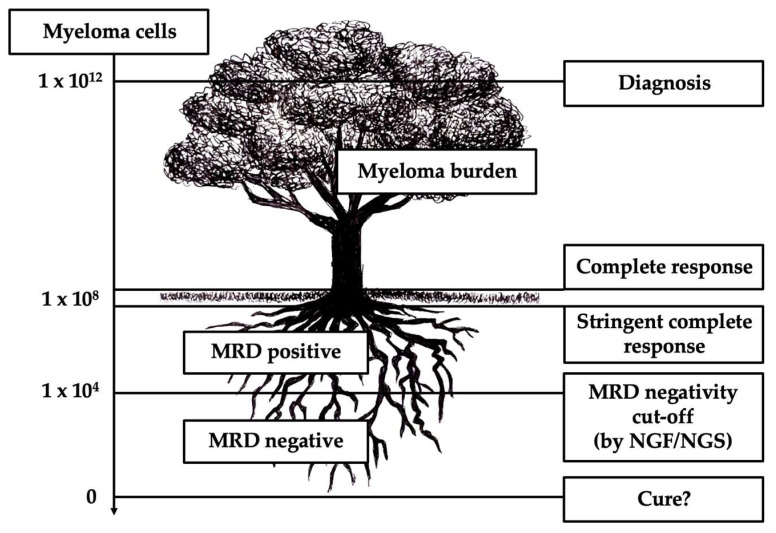
Multiple myeloma disease burden correlated with MRD response (adapted from [58]).

**Table 1 pharmaceuticals-18-00159-t001:** MRD assessment in anti-CD38 MM trials.

Trial	Population	Treatment Scheme	MRD Technique and Sensitivity	MRD Timepoint Assessment	MRD Negativity	MRD Negativity in Patients Achieving ≥ CR
CASSIOPEIA [69]	NDMM TE	VTd vs. Dara-VTd	NGF 10^−5^	100 days after ASCT	44% vs. 64%	20% vs. 34%
PERSEUS [70]	NDMM TE	VRd vs. Dara-VRd, then lenalidomide vs. daratumumab and lenalidomide	NGS 10^−5^ and 10^−6^	After consolidation in >VGPR, then at 12–18–24–36 months	75.2% vs. 47.5%	NA
GRIFFIN [51]	NDMM TE	VRd vs. Dara-VRd	NGS 10^−5^	Once CR or sCR is achieved, at end of induction and consolidation, then after 12 and 24 months of maintenance	20.4% vs. 51%	32.2% vs. 62%
MASTER [71]	NDMM TE	Dara-KRd, then lenalidomide if MRD+	NGS 10^−5^	After induction, 60–80 days after ASCT, after cycles 6 and 10, then after 6 and 18 months	42% after induction; 81% after consolidation	NA
LCI-HEM-MYE-KRdD-001 [72]	NDMM TE or TIE	Dara-KRd, then lenalidomide/observation (A: MRD−), ASCT (B: MRD+ TE), or KRd (C: MRD+ TIE)	NGS and NGF 10^−5^ and 10^−6^	After induction in ≥VGPR, then (A) every 12 months (B) after ASCT (C) every 4 cycles	59% after induction (NGS 10^−5^); (B) 62.5% after ASCT (NGS 10^−5^); (C) 0% (NGS 10^−5^)	NA
GMMG-HD7 [73]	NDMM TE	VRd vs. Isa-VRd, then lenalidomide or isatuximab and lenalidomide	NGF 10^−5^	After induction and ASCT, then every 12 months until end of maintenance	47.7% vs. 66.2% (after ASCT)	25.8% vs. 38.1% (after ASCT)
MAIA [74]	NDMM TIE	Rd vs. Dara-Rd	NGS 10^−5^	Once CR or sCR is achieved, then 12, 18, 24, and 30 months after first dose	10% vs. 31%	34% vs. 58.2%
ALCYONE [75]	NDMM TIE	VMP vs. Dara-VMP	NGS 10^−5^	Once CR or sCR is achieved, then 12, 18, 24 and 30 months after first dose	7% vs. 28.3%	27.8% vs. 58.8%
GMMG-CONCEPT [76]	NDMM TE or TIE	Isa-KRd, then Isa-KR	NGF 10^−5^	After consolidation, then every 6 months	81.8% TE 69.2% TIE	NA
BENEFIT [77]	NDMM TIE	Isa-Rd vs. Isa-VRd	NGS 10^−5^ and 10^−6^	At 12 and 18 months	26% vs. 53% (18 months; cut-off 10^−5^)	17% vs. 37% (18 months; cut-off 10^−5^)
IMROZ [78]	NDMM TIE	VRd vs. Isa-VRd	NGS 10^−5^	At 6, 12, 18, 24, and 36 months in ≥VGPR	44% vs. 58%	40.9% vs. 55.5%
AURIGA [79]	NDMM TE	Lenalidomide vs. daratumumab and lenalidomide in ≥VGPR/MRD+	NGS 10^−5^ and 10^−6^	At 12, 18, 24, and 36 months	18.8% vs. 50.5% (12 months; cut-off 10^−5^)	NA
DART4MM [80]	NDMM TE or TIE	Daratumumab in ≥VGPR/MRD+	NGF 10^−6^	0–6–12–18–24 months	30% MRD conversion	NA
POLLUX [81]	RRMM	Rd vs. Dara-Rd	NGS 10^−5^	Once CR or sCR is achieved, 3 and 6 months following, then every 12 months	6.7% vs. 32.2%	29.2% vs. 57.4%
CASTOR [82]	RRMM	Vd vs. Dara-Vd	NGS 10^−5^	Once CR or sCR is achieved, on cycles 9 and 15, then every 12 months	1.6% vs. 15.1%	17.4% vs. 52.8%
CANDOR [83]	RRMM	Kd vs. Dara-Kd	NGS 10^−5^	At 12 months and at any time	9.1% vs. 27.9%	7.8% vs. 21.8%
APOLLO [84]	RRMM	Pd vs. Dara-Pd	NGS 10^−5^	At time of suspected CR or sCR; at 6, 12, 18, and 24 months; and every 12 months after achieving CR or sCR, until PD	2% vs. 9%	NA
ICARIA-MM [85,86]	RRMM	Pd vs. Isa-Pd	NGS 10^−5^	Once CR or sCR is achieved and 3 months later in case of MRD+	0 vs. 5%	NA
IKEMA [87]	RRMM	Kd vs. Isa-Kd	NGS 10^−5^	On day 1 of every cycle and when treatment stopped	13% vs. 30%	11% vs. 20%

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
