# Peer review of "Minimal Residual Disease Significance in Multiple Myeloma Patients Treated with Anti-CD38 Monoclonal Antibodies"

_pharmaceuticals, 2025, doi:10.3390/ph18020159_

Round 1

Reviewer 1 Report

Comments and Suggestions for Authors

Caroni et al. present a relevant review of MRD results from clinical trials involving anti-CD38 monoclonal antibodies in multiple myeloma. However, the manuscript's language could be improved, and additional references are needed. Some sentences are overly long and complex, making them difficult to read; these should be simplified into shorter, clearer sentences. Additionally, there is inconsistent use of capitalization of drug names and abbreviations throughout.

1.      Provide a proper introduction to both FDA/EMA approved anti-CD38 monoclonal antibodies dara and isa, highlighting their similarities and differences including approved indications. It seems odd that isatuximab is suddenly introduced as “a second-generation monoclonal antibody” at the end of the MRD results in section 2.3.

2.      Provide the methodology used for selecting studies in this review, as some relevant studies appear to be missing (list is not extensive):

a)      Mai EK, et al.; German-Speaking Myeloma Multicenter Group (GMMG) HD7 Investigators; Isatuximab, Lenalidomide, Bortezomib, and Dexamethasone Induction Therapy for Transplant-Eligible Newly Diagnosed Multiple Myeloma: Final Part 1 Analysis of the GMMG-HD7 Trial. J Clin Oncol. 2024 Dec 9:JCO2402266. doi: 10.1200/JCO-24-02266. Epub ahead of print. PMID: 39652594.

b)      Facon T, et al; IMROZ Study Group. Isatuximab, Bortezomib, Lenalidomide, and Dexamethasone for Multiple Myeloma. N Engl J Med. 2024 Oct 31;391(17):1597-1609. doi: 10.1056/NEJMoa2400712. Epub 2024 Jun 3. PMID: 38832972.

c)      Leleu X, et al. Isatuximab, lenalidomide, dexamethasone and bortezomib in transplant-ineligible multiple myeloma: the randomized phase 3 BENEFIT trial. Nat Med. 2024 Aug;30(8):2235-2241. doi: 10.1038/s41591-024-03050-2. Epub 2024 Jun 3. PMID: 38830994; PMCID: PMC11333283.

d)      Leypoldt LB, Tichy D, Besemer B, Hänel M, Raab MS, Mann C, Munder M, Reinhardt HC, Nogai A, Görner M, Ko YD, de Wit M, Salwender H, Scheid C, Graeven U, Peceny R, Staib P, Dieing A, Einsele H, Jauch A, Hundemer M, Zago M, Požek E, Benner A, Bokemeyer C, Goldschmidt H, Weisel KC. Isatuximab, Carfilzomib, Lenalidomide, and Dexamethasone for the Treatment of High-Risk Newly Diagnosed Multiple Myeloma. J Clin Oncol. 2024 Jan 1;42(1):26-37. doi: 10.1200/JCO.23.01696. Epub 2023 Sep 27. PMID: 37753960; PMCID: PMC10730063.

e)      Bhutani M, et al. MRD-driven phase 2 study of daratumumab, carfilzomib, lenalidomide and dexamethasone in newly diagnosed multiple myeloma. Blood Adv. 2024 Nov 22:bloodadvances.2024014417. doi: 10.1182/bloodadvances.2024014417. Epub ahead of print. PMID: 39576965.

3.      Consider adding subheadings for Section 2.3 and dividing it into logical subsections, such as: ND-MM, transplant-eligible; ND-MM, transplant-ineligible; RR-MM, prior IMiD; RR-MM, prior PI, etc.

4.      In RR-MM trials, specify the prior treatments that patients received before enrollment. This information is important for the interpretation of the MRD response.

5.      Section 3 lacks a true discussion of presented studies.

6.      In the future discussion section, clarify the sensitivity of new methodologies compared to existing methods, particularly when using terms like 'highly sensitive'.

7.      Include page numbers for all references listed in the reference list.

8.      Provide citations for the studies listed in Table 1.

9.      Avoid citing meeting abstracts and full papers of the same study together unless the meeting abstract provides additional information not found in the full paper. For example, in Line 129, Reference 70 is the meeting abstract for full paper Reference 69 but does not contain the specific information stated in the text (median PFS from the second randomization). Same with references 83/84, 85/86 and other.

10.  Provide a reference for the disease burden numbers shown in Figure 1.

11.  This should be obvious: Explain all abbreviations when they first appear in the text and consistently use them. For example, in Line 107: “In the NDMM setting…” should be written as “In the newly diagnosed (ND) MM setting. This rule also applies to the table—provide explanations for all abbreviations.

12.  Define HRCA (del(17p), t(4;14) etc.) and “SR”.

13.  Clearly specify the phase of all mentioned clinical trials, as some involve similar treatments in comparable populations (e.g., GRIFFIN trial, Phase 2, vs. PERSEUS trial, Phase 3).

14.  The relevance of the final paragraph of Section 2.3 is unclear, and it appears rather long. Is this the only MRD analysis in patients without anti-CD38 treatment? Why is the MRD-negativity rate for each arm not mentioned?

15.  Cite references after the first relevant sentence. For instance, cite reference [73] after “A combined analysis of all pts enrolled in POLLUX, CASTOR, MAIA and ALCYONE trials was recently conducted in order to evaluate the rate of MRD-negativity on a large cohort of pts enrolled in pivotal studies.”

16.  Write the names of all agents in small caps, and always use the same abbreviations of these agents.

17.  Write “patients” instead of “pts”.

18.  Decimal points should be represented by periods, not commas (for example, Line 244: “19,15 months”).

19.  Are both references 80 and 81 essential (same trial)?

20.  Line 9: "to measure the deepness of response and novel drug efficacy" should be "to measure the depth of response and novel drug efficacy."

21.  Line 17: "MRD driven therapy seems appealing for the future of MM patients" should be "MRD-driven therapy appears promising for the future of MM patients".

22.  Line 25: “has been showed” should be “has been shown”. Furthermore, consider replacing “to produce damage” with “to cause harm”.

23.  Line 30: Consider rephrasing “Old drugs before early year 2000 were represented only by chemotherapy”.

24.  Line 35: "...drugs have been used alone or with others in triplet and quadruplet..." should be "...drugs have been used alone or in combination with others in triplets and quadruplets".

25.  Line 45-47: “The percentage of MRD negativity reached by anti CD38 MoA Daratumumab and Isatuximab has been unprecedented in multiple myeloma.” should be something like “The rate of MRD negativity reached by combination therapies including anti-CD38 MoA daratumumab or isatuximab have been unprecedented in multiple myeloma.”

26.  Line 56: "contribute to discriminating normal from clonal MM PCs" should be "contribute to distinguishing normal from clonal MM PCs".

27.  Line 65: "a 107 cell/sample" should be "107 cells per sample".

28.  Line 78: "...and required expertise for data analysis" should be "...and the expertise required for data analysis".

29.  Line 88: "come" should be "came".

30.  Line 91: Provide a reverence.

31.  Line 95: "electrophoresis" should be "serum protein electrophoresis".

32.  Line 99: "o near-CR" should be "or near-CR.

33.  Line 122: “...the incidence of MRD-negativity with Dara-VTd evaluated by NGF cytometry was 64% and 57% by NGS...” should be “...the rate of MRD-negativity with Dara-VTd, evaluated using NGF cytometry, was 64%, and 57% using NGS...”.

34.  Line 133: "VRD" should be "VRd".

35.  Line 138-139: Consider rephrasing “MRD negative patients were 75,2% vs 47,5%, respectively” – the wording is unclear. For example: “MRD negativity was observed in 75.2% of the treatment group vs. 47.5% of the control group, respectively”. Further, decimal points should be represented by periods, not commas.

36.  Line 122 and others: "incidence" should be "rate" ("incidence" refers to number of new cases of a disease or event in a population over a specific period, while "rate" is more general).

37.  Line 144: "response rates deepened" should be "response rates increased".

38.  Line 158: “progressive disease” should be “progressive disease (PD)”.

39.  Line 162: "...MRD was evaluated in NGS..." should be "...MRD was evaluated using/by NGS…".

40.  Line 169: "monthly-Daratumumab maintenance" should be "monthly daratumumab maintenance".

41.  Line 206: "achieved a CR MRD-negative" should probably be "achieved a CR and were MRD-negative".

42.  Line 217: "could not detect a measurable MRD" should be "could not detect MRD".

43.  Line 219: "have been found" should be "were found".

44.  Line 228: What is “suspected CR”?

45.  Line 245: “sensitivity of 1 in 105 nucleated cells” should be “sensitivity of 10−5 nucleated cells” for consistent use of language.

46.  Line 251: "MRD negative conversion" should be "MRD negativity conversion rate".

47.  Line 259: Provide reference for quality-of-life improvement through treatment with anti-CD38 MoA.

48.  Line 259:"Bone disease should be also evaluated at diagnosis and to confirm response with metabolic imaging techniques" should be "Bone disease should also be evaluated at diagnosis and for confirming response using metabolic imaging techniques".

49.  Line 263: “detection circulating tumor cells” should be “detections of circulating tumor cells”.

50.  Line 265-266: “Recent advances in flow cytometry led to the identification of CTCs, thus offering a novel potential method to assess MRD as a liquid biopsy [99].” Sentence seems to be redundant.

51.  Line 266-270: “One study demonstrated that positive MRD in PB by detection of CTCs could be a surrogate of MRD in BM and represent an independent prognostic factor, however, the high rate of false negatives in PB indicates that additional studies are required to better define the role of MRD assessment by CTCs [100].” – This should be explained with more details.

52.   Line 275: Why is MS an “attractive” method compared to other methods?

53.  Line 277-278: "that MS should be combined with tradition MRD" should be "that MS could be combined with NGS or NGF MRD"

54.  Line 286-296: There are several changes needed due to incorrect use of English language:

a)       "With an MRD-oriented attitude" → "With an MRD-oriented approach"

b)      "2 years after the start of maintenance therapy pts discontinued the treatment" → "Patients discontinued maintenance therapy 2 years after its initiation"

c)      "if they resulted MRD-negative" → "if they were found to be MRD-negative"

d)      "evaluated through NGF cytometry on bone marrow blood with a sensitivity 3x106" → "as assessed by NGF cytometry on bone marrow [aspirate] with a sensitivity of 3×10⁶"

e)      "at the opposite" → "Conversely"

f)       "carried on the maintenance therapy" → "continued maintenance therapy"

g)      "further" → "an additional"

h)      "66-months PFS" → "66-month PFS"

55.  Line 288: "blood" should be "aspirate".

56.  Line 298: "have been found" should be "were found".

57.  Line 303: "towards a cure" should be "toward a cure".

Comments on the Quality of English Language

See above.

Author Response

Answers to reviewer 1

Comments and Suggestions for Authors

Caroni et al. present a relevant review of MRD results from clinical trials involving anti-CD38 monoclonal antibodies in multiple myeloma. However, the manuscript's language could be improved, and additional references are needed. Some sentences are overly long and complex, making them difficult to read; these should be simplified into shorter, clearer sentences. Additionally, there is inconsistent use of capitalization of drug names and abbreviations throughout. 

  1. Provide a proper introduction to both FDA/EMA approved anti-CD38 monoclonal antibodies dara and isa, highlighting their similarities and differences including approved indications. It seems odd that isatuximab is suddenly introduced as “a second-generation monoclonal antibody” at the end of the MRD results in section 2.3.
  2. We introduced daratumumab and isatuximab with their current indications.
  3. Provide the methodology used for selecting studies in this review, as some relevant studies appear to be missing (list is not extensive): 
  4. a)Mai EK, et al.; German-Speaking Myeloma Multicenter Group (GMMG) HD7 Investigators; Isatuximab, Lenalidomide, Bortezomib, and Dexamethasone Induction Therapy for Transplant-Eligible Newly Diagnosed Multiple Myeloma: Final Part 1 Analysis of the GMMG-HD7 Trial. J Clin Oncol. 2024 Dec 9:JCO2402266. doi: 10.1200/JCO-24-02266. Epub ahead of print. PMID: 39652594.
  5. b)Facon T, et al; IMROZ Study Group. Isatuximab, Bortezomib, Lenalidomide, and Dexamethasone for Multiple Myeloma. N Engl J Med. 2024 Oct 31;391(17):1597-1609. doi: 10.1056/NEJMoa2400712. Epub 2024 Jun 3. PMID: 38832972.
  6. c)Leleu X, et al. Isatuximab, lenalidomide, dexamethasone and bortezomib in transplant-ineligible multiple myeloma: the randomized phase 3 BENEFIT trial. Nat Med. 2024 Aug;30(8):2235-2241. doi: 10.1038/s41591-024-03050-2. Epub 2024 Jun 3. PMID: 38830994; PMCID: PMC11333283. 
  7. d)Leypoldt LB, Tichy D, Besemer B, Hänel M, Raab MS, Mann C, Munder M, Reinhardt HC, Nogai A, Görner M, Ko YD, de Wit M, Salwender H, Scheid C, Graeven U, Peceny R, Staib P, Dieing A, Einsele H, Jauch A, Hundemer M, Zago M, Požek E, Benner A, Bokemeyer C, Goldschmidt H, Weisel KC. Isatuximab, Carfilzomib, Lenalidomide, and Dexamethasone for the Treatment of High-Risk Newly Diagnosed Multiple Myeloma. J Clin Oncol. 2024 Jan 1;42(1):26-37. doi: 10.1200/JCO.23.01696. Epub 2023 Sep 27. PMID: 37753960; PMCID: PMC10730063.
  8. e)Bhutani M, et al. MRD-driven phase 2 study of daratumumab, carfilzomib, lenalidomide and dexamethasone in newly diagnosed multiple myeloma. Blood Adv. 2024 Nov 22:bloodadvances.2024014417. doi: 10.1182/bloodadvances.2024014417. Epub ahead of print. PMID: 39576965.
  9. We added the listed studies as suggested. We did not mention them because we focused on trials who led to EMA’s approval of daratumumab and isatuximab
  10. Consider adding subheadings for Section 2.3 and dividing it into logical subsections, such as: ND-MM, transplant-eligible; ND-MM, transplant-ineligible; RR-MM, prior IMiD; RR-MM, prior PI, etc. 
  11. We did a synthetic partition of each trial’s population setting (NDMM, TIE or TE, or RRMM) in table 1 and we added an introduction to section 2.3.
  12. In RR-MM trials, specify the prior treatments that patients received before enrollment. This information is important for the interpretation of the MRD response.
  13. In CASTOR trial the median number of prior lines of therapy was 2 and more than half of the patients were previously treated with bortezomib and/or IMIDs. In POLLUX trial the median number of prior lines of therapy was 1, more than half of the patients were previously treated with IMIDs and 85% with bortezomib. In APOLLO trial the median number of prior lines of therapy was 2 and all the patients were previously treated with IMIDs and bortezomib. In IKEMA trial the median number of prior lines of therapy was 2 and more than half of the patients were previously treated with IMIDs and PI, only 4% were treated with other MoA therapy. In ICARIA-MM trial the median number of prior lines of therapy was 3 and all the patients were previously treated with PIs and lenalidomide.

We added these data but please consider that, we were encouraged to shorten the description of trials by the other reviewer.

  1. Section 3 lacks a true discussion of presented studies. 

5 The studies were extensively described in section 2. If the importance of MRD in MM was stressed by the results of the presented trials, the aim of section 3 was to discuss what the future could reserve for MRD, in terms of new possible techniques for MRD evaluation and about MRD as a tool for establishing intensification, de-escalation or discontinuation of therapies.

In the future discussion section, clarify the sensitivity of new methodologies compared to existing methods, particularly when using terms like 'highly sensitive'.

  1. We reported on new MRD techniques that are under evaluation. However, these methods have not been validated by official guidelines and are not routinely used in clinical practice. Extensive comparison between methodologies is not the purpose of our review, which is focused on the current knowledge of MRD value in MM.
  2. Include page numbers for all references listed in the reference list.
  3. We modified as requested.
  4. 8.Provide citations for the studies listed in Table 1. 
  5. We added citations as requested.
  6. Avoid citing meeting abstracts and full papers of the same study together unless the meeting abstract provides additional information not found in the full paper. For example, in Line 129, Reference 70 is the meeting abstract for full paper Reference 69 but does not contain the specific information stated in the text (median PFS from the second randomization). Same with references 83/84, 85/86 and other.

9) We reformulated many sentences and shortened the description of trials, hence we removed some useless citations.

  1. Provide a reference for the disease burden numbers shown in Figure 1. 
  2. We added the reference.
  3. This should be obvious: Explain all abbreviations when they first appear in the text and consistently use them. For example, in Line 107: “In the NDMM setting…” should be written as “In the newly diagnosed (ND) MM setting. This rule also applies to the table—provide explanations for all abbreviations. 
  4. We modified as requested.
  5. Define HRCA (del(17p), t(4;14) etc.) and “SR”.
  6. We specified which cytogenetic abnormalities were considered high risk in the mentioned studies. Del(17p), t(4;14) and t(4;16) were considered HR in MASTER trial, in the metanalysis by Munshi et al. (citation 67) and in the combined analysis of patients enrolled in POLLUX, CASTOR, MAIA and ALCYONE trials by Cavo et al. (citation 72). Only in GMMG-CONCEPT also gain(1q) was included among HRCA, as reported.
  7. Clearly specify the phase of all mentioned clinical trials, as some involve similar treatments in comparable populations (e.g., GRIFFIN trial, Phase 2, vs. PERSEUS trial, Phase 3). 
  8. We specified as requested.
  9. The relevance of the final paragraph of Section 2.3 is unclear, and it appears rather long. Is this the only MRD analysis in patients without anti-CD38 treatment? Why is the MRD-negativity rate for each arm not mentioned? 
  10. The MRD negativity rates for both treatment arms in AURIGA trial was reported. The MRD negativity rate in DART4MM trial was mentioned too, since it is a sigle arm trial.
  11. Cite references after the first relevant sentence. For instance, cite reference [73] after “A combined analysis of all pts enrolled in POLLUX, CASTOR, MAIA and ALCYONE trials was recently conducted in order to evaluate the rate of MRD-negativity on a large cohort of pts enrolled in pivotal studies.” 
  12. We modified as requested.
  13. Write the names of all agents in small caps, and always use the same abbreviations of these agents.
  14. We modified as requested.
  15. Write “patients” instead of “pts”.
  16. We modified as requested.
  17. Decimal points should be represented by periods, not commas (for example, Line 244: “19,15 months”).
  18. We corrected commas into periods as requested.
  19. Are both references 80 and 81 essential (same trial)?
  20. No, we removed the first reference.
  21. Line 9: "to measure the deepness of response and novel drug efficacy" should be "to measure the depth of response and novel drug efficacy." 
  22. We modified as requested.
  23. Line 17: "MRD driven therapy seems appealing for the future of MM patients" should be "MRD-driven therapy appears promising for the future of MM patients". 
  24. We modified as requested.
  25. Line 25: “has been showed” should be “has been shown”. Furthermore, consider replacing “to produce damage” with “to cause harm”. 
  26. We modified as requested.
  27. Line 30: Consider rephrasing “Old drugs before early year 2000 were represented only by chemotherapy”.
  28. We rephrased as requested.
  29. Line 35: "...drugs have been used alone or with others in triplet and quadruplet..." should be "...drugs have been used alone or in combinationwith others in triplets and quadruplets". 
  30. We modified as requested.
  31. Line 45-47: “The percentage of MRD negativity reached by anti CD38 MoA Daratumumab and Isatuximab has been unprecedented in multiple myeloma.” should be something like “The rate of MRD negativity reached by combination therapies including anti-CD38 MoA daratumumab or isatuximab have been unprecedented in multiple myeloma.” 
  32. We modified as requested.
  33. Line 56: "contribute to discriminating normal from clonal MM PCs" should be "contribute to distinguishing normal from clonal MM PCs". 
  34. We modified as requested.
  35. Line 65: "a 107cell/sample" should be "107 cells per sample". 
  36. We modified as requested.
  37. Line 78: "...and required expertise for data analysis" should be "...and theexpertise required for data analysis". 
  38. We modified as requested.
  39. Line 88: "come" should be "came". 
  40. We modified as requested.
  41. Line 91: Provide a reverence.
  42. The reference is #66.
  43. Line 95: "electrophoresis" should be "serum protein electrophoresis". 
  44. We modified as requested.
  45. Line 99: "o near-CR" should be "or near-CR. 
  46. We modified as requested.
  47. Line 122: “...the incidence of MRD-negativity with Dara-VTd evaluated by NGF cytometry was 64% and 57% by NGS...” should be “...the rate of MRD-negativity with Dara-VTd, evaluated using NGF cytometry, was 64%, and 57% using NGS...”.
  48. We modified as requested.
  49. Line 133: "VRD" should be "VRd".
  50. The entire sentence has been reformulated.
  51. Line 138-139: Consider rephrasing “MRD negative patients were 75,2% vs 47,5%, respectively” – the wording is unclear. For example: “MRD negativity was observed in 75.2% of the treatment group vs. 47.5% of the control group, respectively”. Further, decimal points should be represented by periods, not commas.
  52. We modified as requested.
  53. Line 122 and others: "incidence" should be "rate" ("incidence" refers to number of new cases of a disease or event in a population over a specific period, while "rate" is more general). 
  54. We corrected as requested and in all similar cases.
  55. Line 144: "response rates deepened" should be "response rates increased". 
  56. We modified as requested.
  57. Line 158: “progressive disease” should be “progressive disease (PD)”. 
  58. We modified as requested.
  59. Line 162: "...MRD was evaluated in NGS..." should be "...MRD was evaluated using/by NGS…". 
  60. We modified as requested.
  61. Line 169: "monthly-Daratumumab maintenance" should be "monthly daratumumab maintenance". 
  62. We modified as requested.
  63. Line 206: "achieved a CR MRD-negative" should probably be "achieved a CR and were MRD-negative". 
  64. We modified as requested.
  65. Line 217: "could not detect a measurable MRD" should be "could not detect MRD". 
  66. We modified as requested.
  67. Line 219: "have been found" should be "were found". 
  68. We modified as requested.
  69. Line 228: What is “suspected CR”? 
  70. According to IMWG response criteria, a bone marrow aspiration or biopsy is required to confirm CR or sCR. Hence, when immunofixation was negative and it was supposed to have obtained at least a CR a bone marrow aspirate was done, including MRD evaluation.
  71. Line 245: “sensitivity of 1 in 105nucleated cells” should be “sensitivity of 10−5 nucleated cells” for consistent use of language.
  72. We modified as requested.
  73. Line 251: "MRD negative conversion" should be "MRD negativity conversion rate". 
  74. We modified as requested.
  75. Line 259: Provide reference for quality-of-life improvement through treatment with anti-CD38 MoA.
  76. We modified the sentence.

  1. Line 259:"Bone disease should be also evaluated at diagnosis and to confirm response with metabolic imaging techniques" should be "Bone disease should also be evaluated at diagnosis and for confirming response using metabolic imaging techniques". 
  2. We modified as requested.
  3. Line 263: “detection circulating tumor cells” should be “detections of circulating tumor cells”. 
  4. We modified as requested.
  5. Line 265-266: “Recent advances in flow cytometry led to the identification of CTCs, thus offering a novel potential method to assess MRD as a liquid biopsy [99].” Sentence seems to be redundant.
  6. We removed the sentence as suggested.
  7. Line 266-270: “One study demonstrated that positive MRD in PB by detection of CTCs could be a surrogate of MRD in BM and represent an independent prognostic factor, however, the high rate of false negatives in PB indicates that additional studies are required to better define the role of MRD assessment by CTCs [100].” – This should be explained with more details.
  8. We rephrased the sentence for a better comprehension.
  9. Line 275: Why is MS an “attractive” method compared to other methods?
  10. We considered MS an “attractive” method because it can reveal residual M protein at very low levels, hence it could be used for MRD detection without needing to perform invasive procedures (i.e. bone marrow aspirate) for NGF/NGS MRD.
  11. Line 277-278: "that MS should be combined with tradition MRD" should be "that MS could be combined with NGS or NGF MRD"
  12. We modified as requested.
  13. Line 286-296: There are several changes needed due to incorrect use of English language:
  14. a)"With an MRD-oriented attitude" → "With an MRD-oriented approach"
  15. b)"2 years after the start of maintenance therapy pts discontinued the treatment" → "Patients discontinued maintenance therapy 2 years after its initiation"
  16. c)"if they resulted MRD-negative" → "if they were found to be MRD-negative"
  17. d)"evaluated through NGF cytometry on bone marrow blood with a sensitivity 3x106" → "as assessed by NGF cytometry on bone marrow [aspirate] with a sensitivity of 3×10⁶"
  18. e)"at the opposite" → "Conversely"
  19. f)"carried on the maintenance therapy" → "continued maintenance therapy"
  20. g)"further" → "an additional"
  21. h)"66-months PFS" → "66-month PFS" 
  22. We modified as suggested.
  23. Line 288: "blood" should be "aspirate".
  24. We modified as requested.
  25. Line 298: "have been found" should be "were found". 
  26. We modified as requested.
  27. Line 303: "towards a cure" should be "toward a cure". 
  28. We modified as requested.

Comments on the Quality of English Language

See above.

Reviewer 2 Report

Comments and Suggestions for Authors

The presented review is devoted to the importance of evaluation of minimal residual disease in multiple myeloma. It summarizes the approaches to minimal residual disease evaluation as well as the clinical trials with minimal residual disease as primary end-points.

The specific comments are below:

1)     Thorough English editing is needed. The text of the paper contains a lot of typos, conversational phases and repeats.

2)     The definition of minimal residual disease should be given in the Introduction.

3)     The review should be better organized, in particular the description of multiple clinical trials should be more concise.

4)     The Abstract should be broadening with the conclusion of the review.

5)     The review contains multiple abbreviations, and their extensive use makes the manuscript difficult to read. As an option, authors could replace the abbreviation of research group or clinical trial with the reference. The abbreviation list should be added.

6)     The Table 1 should be moved to the beginning of Section “2.3 MRD and trials with anti-CD38 monoclonal antibodies”.

7)     The information how MRD was followed in clinical trials should be added in the Section “2.2 MRD initial experience in clinical trials”.

Comments on the Quality of English Language

 Thorough English editing is needed. The text of the paper contains a lot of typos, conversational phases and repeats.

Author Response

Answer to reviewer 2

Comments and Suggestions for Authors

The presented review is devoted to the importance of evaluation of minimal residual disease in multiple myeloma. It summarizes the approaches to minimal residual disease evaluation as well as the clinical trials with minimal residual disease as primary end-points. 

The specific comments are below:

1)     Thorough English editing is needed. The text of the paper contains a lot of typos, conversational phases and repeats.

1) We reviewed the English as requested.

2)     The definition of minimal residual disease should be given in the Introduction.

2) We provided the definition of minimal residual disease in the introduction as requested.

3)     The review should be better organized, in particular the description of multiple clinical trials should be more concise.

3) We shortened the description of trials as requested.

4)     The Abstract should be broadening with the conclusion of the review.

4) We modified the abstract as requested.

5)     The review contains multiple abbreviations, and their extensive use makes the manuscript difficult to read. As an option, authors could replace the abbreviation of research group or clinical trial with the reference. The abbreviation list should be added.

5) We reduced abbreviations as requested. Abbreviations list was added.

6)     The Table 1 should be moved to the beginning of Section “2.3 MRD and trials with anti-CD38 monoclonal antibodies”.

6) We moved table 1 as requested.

7)     The information how MRD was followed in clinical trials should be added in the Section “2.2 MRD initial experience in clinical trials”.

7) We added information as requested

Comments on the Quality of English Language

Thorough English editing is needed. The text of the paper contains a lot of typos, conversational phases and repeats.

Round 2

Reviewer 1 Report

Comments and Suggestions for Authors

The manuscript is greatly improved!

1.      The relationship between MRD depth and clinical response (CR, sCR = 10E-8), as shown in Figure 1, is not supported by reference #61.

2.      ll. 37: “peculiar” doesn’t seem right here.

3.      ll. 41: “Since responses are increased […]” -> “Since response rates have increased, MRD was introduced by the International Myeloma Working Group (IMWG) [for] response evaluation.”

4.      ll. 51-53: This sentence is a bit convoluted and needs revision.

5.      ll. 54: “combinations therapies” -> “combination therapies”

6.      ll. 66-68: ‘for’ instead of ‘from’: “Isatuximab is approved for the second line in combination with carfilzomib and dexamethasone or for the third line with pomalidomide and dexamethasone.”

7.      ll. 135: “randomized [to]”

8.      ll. 156: “associated [with]”

9.      ll. 158-160: “For what concerns MRD-negativity rates, they resulted higher […]” – This sentence is a bit awkward.

10.  Some remaining minor punctuation errors.

Author Response

The manuscript is greatly improved! thank you for your useful suggestions

  1. The relationship between MRD depth and clinical response (CR, sCR = 10E-8), as shown in Figure 1, is not supported by reference #61.

We substituted with another reference

  1. 37: “peculiar” doesn’t seem right here.

We erased it

  1. 41: “Since responses are increased […]” -> “Since response rates have increased, MRD was introduced by the International Myeloma Working Group (IMWG) [for] response evaluation.”

We substituted accordingly

  1. 51-53: This sentence is a bit convoluted and needs revision.

We modified as suggested thanks

  1. 54: “combinations therapies” -> “combination therapies”

yes, thank you

  1. 66-68: ‘for’ instead of ‘from’: “Isatuximab is approved for the second line in combination with carfilzomib and dexamethasone or for the third line with pomalidomide and dexamethasone.”

Yes thank you

  1. ll. 135: “randomized [to]”yes thanks
  2. ll. 156: “associated [with]” yes
  3. ll. 158-160: “For what concerns MRD-negativity rates, they resulted higher […]” – This sentence is a bit awkward.

We agree and we modified as suggested

  1. Some remaining minor punctuation errors.

we went through the manuscript and tried to correct all, thanks

Reviewer 2 Report

Comments and Suggestions for Authors

No further comments

Author Response

thank you